# Conductance through a helical state in an Indium antimonide nanowire

J. Kammhuber[1], M.C. Cassidy [1], F. Pei[1], M.P. Nowak[1,2], A. Vuik[1], Ö. Gül[1], D. Car[1,3], S.R. Plissard [4], E.P.A.M. Bakkers [1,3], M. Wimmer[1] & L.P. Kouwenhoven[1]

The motion of an electron and its spin are generally not coupled. However in a one-dimensional material with strong spin-orbit interaction (SOI) a helical state may emerge at finite magnetic fields, where electrons of opposite spin will have opposite momentum. The existence of this helical state has applications for spin filtering and cooper pair splitter devices and is an essential ingredient for realizing topologically protected quantum computing using Majorana zero modes. Here, we report measurements of a quantum point contact in an indium antimonide nanowire. At magnetic fields exceeding 3 T, the $2\,e^2/h$ conductance plateau shows a re-entrant feature toward $1\,e^2/h$ which increases linearly in width with magnetic field. Rotating the magnetic field clearly attributes this experimental signature to SOI and by comparing our observations with a numerical model we extract a spin-orbit energy of approximately 6.5 meV, which is stronger than the spin-orbit energy obtained by other methods.

---

[1] QuTech and Kavli Institute of Nanoscience, Delft University of Technology, Delft 2600 GA, The Netherlands. [2] Faculty of Physics and Applied Computer Science, AGH University of Science and Technology, al. A. Mickiewicza 30, Kraków 30-059, Poland. [3] Department of Applied Physics, Eindhoven University of Technology, Eindhoven 5600 MB, The Netherlands. [4] CNRS-Laboratoire d'Analyse et d'Architecture des Systèmes (LAAS), Université de Toulouse, 7 avenue du colonel Roche, Toulouse F-31400, France. Correspondence and requests for materials should be addressed to L.P.K. (email: L.P.Kouwenhoven@tudelft.nl)

The spin-orbit interaction (SOI) is a relativistic effect where a charged particle moving in an electric field $E$ with momentum $k$ and velocity $v = k/m_0$, experiences an effective magnetic field $B_{SO} = (-1/m_0 c)k \times E$ in its rest frame. The magnetic moment of the electron spin, $\mu = eS/m_0$, interacts with this effective magnetic field, resulting in a spin-orbit Hamiltonian $H_{SO} = -\mu.B_{SO}$ that couples the spin to the orbital motion and electric field. In crystalline materials, the electric field arises from a symmetry breaking that is either intrinsic to the underlying crystal lattice in which the carriers move, known as the Dresselhaus SOI[1], or an artificially induced asymmetry in the confinement potential due to an applied electric field, or Rashba[2] SOI. Wurtzite and certain zincblende nanowires possess a finite Dresselhaus SOI, and so the SOI is a combination of both the Rashba and Dresselhaus components. For zincblende nanowires grown along the [111] growth direction the crystal lattice is inversion symmetric, and so only a Rashba component to the spin-orbit interaction is thought to remain[3].

Helical states[4, 5] have been shown to emerge in the edge mode of two-dimensional (2D) quantum spin hall topological insulators[6, 7], and in quantum wires created in GaAs cleaved edge overgrowth samples[8]. They have also been predicted to exist in carbon nanotubes under a strong applied electric field[9], RKKY systems[10], and in InAs and InSb semiconducting nanowires where they are essential for the formation of Majorana zero modes[11–13]. Although the signatures of Majoranas have been observed in nanowire-superconductor hybrid devices[14, 15], explicit demonstration of the helical state in these nanowires has remained elusive. The measurement is expected to show a distinct experimental signature of the helical state—a return to $1\,e^2/h$ conductance at the $2\,e^2/h$ plateau in increasing magnetic field as different portions of the band dispersion are probed[4, 5, 16]. While ballistic transport through nanowire quantum point contacts (QPCs) is now standard[17, 18], numerical simulations have shown that the visibility of this experimental signature critically depends on the exact combination of geometrical and physical device parameters[16].

Here, we observe a clear signature of transport through a helical state in a QPC formed in an InSb nanowire when the magnetic field has a component perpendicular to the spin-orbit field. We show that the state evolves under rotation of the external magnetic field, disappearing when the magnetic field is aligned with $B_{SO}$. By comparing our data to a theoretical model, we extract a spin-orbit energy $E_{SO} = 6.5$ meV, significantly stronger than that measured in InSb nanowires by other techniques.

## Results

**Emergence of the helical gap a quantum point contact**. Figure 1a shows a schematic image of a typical QPC device. An InSb nanowire is deposited on a degenerately doped silicon wafer covered with a thin (20 nm) SiN dielectric. The QPC is formed in the nanowire channel in a region defined by the source and drain contacts spaced ~325 nm apart. The chemical potential $\mu$ in the

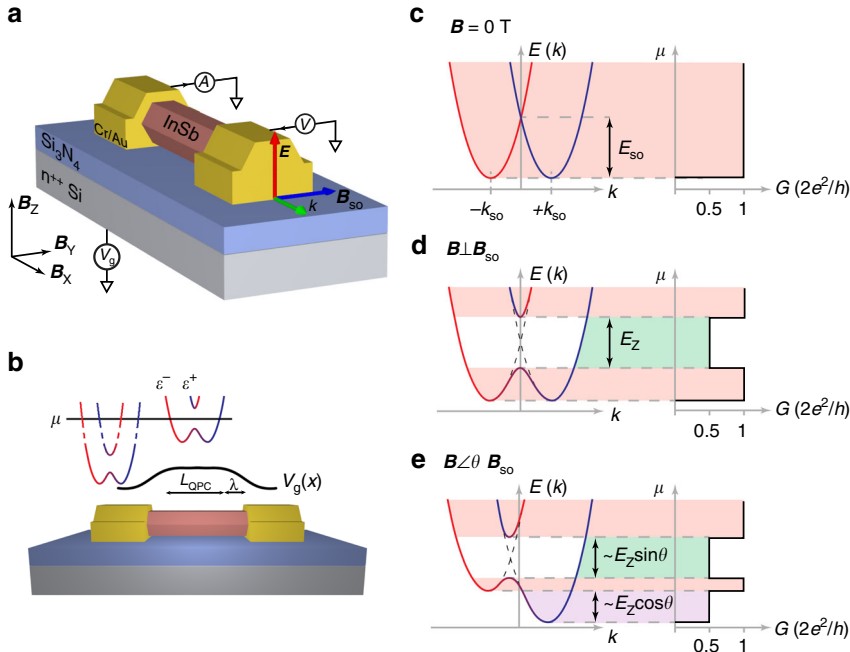

**Fig. 1** The helical gap in a one-dimensional nanowire device. **a** An indium antimonide (InSb) nanowire device with a Rashba spin-orbit field $B_{SO}$ perpendicular to the wave vector $k$ and the electric field $E$. A voltage is sourced to one contact, and the resulting conductance measured from the second contact. A degenerately doped wafer acts as global backgate $V_g$. **b** A quantum point contact (QPC) of length $L$ is defined by the two contacts. Underneath the nanowire contacts, many subbands are occupied as the contacts screen the gate electric field. In the nanowire channel away from the contacts, the chemical potential in the wire, $\mu$, is tuned with $V_g$. The onset shape of $V_g$ with a lengthscale $\lambda$ is set by the dielectric and screening of the electric field from the metallic contacts resulting in an effective QPC length $L_{QPC} = L - 2\lambda$. **c** The energy dispersion of the first two subbands for a system with spin-orbit interaction (SOI) at external magnetic field $B = 0$ T. The SOI causes subbands to shift by $k_{SO}$ in momentum space, as electrons with opposite spins carry opposite momentum. When the electrochemical potential $\mu$ in the wire is tuned conductance plateaus will occur at integer values of $G_0$. **d** At finite magnetic field $B$ perpendicular to $B_{SO}$, the spin polarized bands hybridize opening a helical gap of size $E_Z$ (green). In this region the conductance reduces from $1 \cdot G_0$ to $0.5 \cdot G_0$ when $\mu$ is positioned inside the gap. **e** When the magnetic field is orientated at an angle $\theta$ to $B_{SO}$, the size of the helical gap decreases to only include the component of the magnetic field perpendicular to $B_{SO}$ and the two subbands split by an additional Zeeman gap (purple). The color scheme illustrating different conductance regimes is also used in Figs. 2d and 3b. For all angles the re-entrant conductance feature at $0.5 \cdot G_0$ in the $1 \cdot G_0$ plateau will scale linearly with Zeeman energy

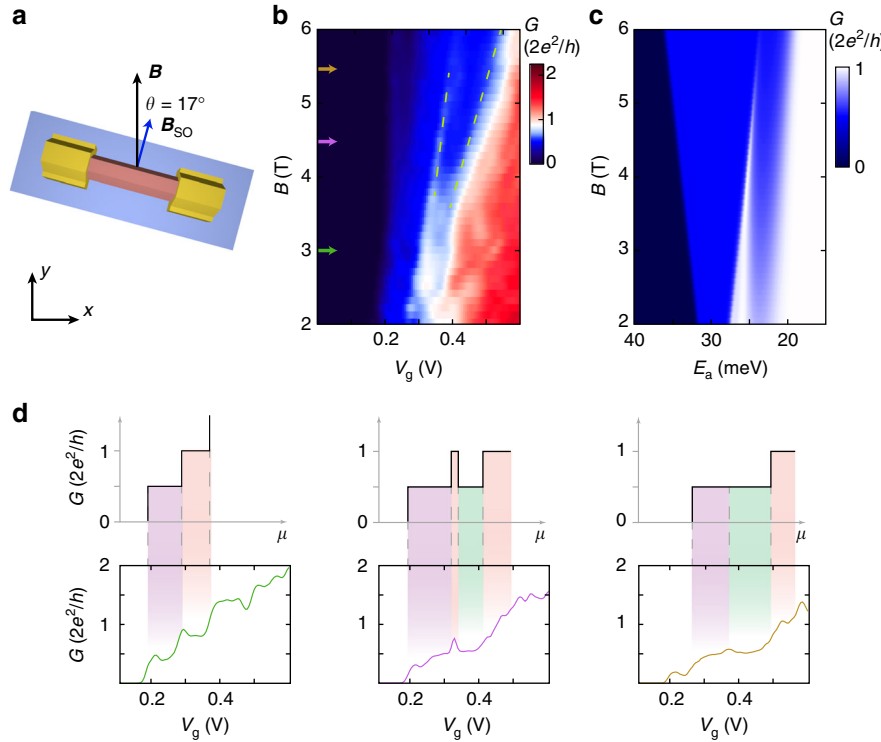

**Fig. 2** Magnetic field dependence of the helical gap. **a** The nanowire lies in the $x$–$y$ plane at an angle $\theta = 17°$ relative to the external magnetic field. **b** Differential conductance d$I$/d$V$ at zero source-drain bias as a function of backgate voltage and external magnetic field. At low magnetic fields conductance plateaus at multiples of $0.5 \cdot G_0$ are visible. Above $B = 3$ T, a re-entrant conductance feature at $0.5 \cdot G_0$ appears in the $1 \cdot G_0$ plateau. The feature evolves linearly with Zeeman energy indicated by dashed green lines. **c** Numerical simulations of the differential conductance as a function of the potential $E_a$ and external magnetic field for $L = 325$ nm, $\theta = 17°$ and $l_{SO} = 20$ nm (Supplementary Note 1 and Supplementary Fig. 1 for a more detailed description of the model). In the numerical simulations, the conductance plateaus have a different slope compared to the experimental data as the calculations neglect screening by charges in the wire. **d** (*Top*) Sketch of the expected conductance together with the color scheme explained in Fig. 1 and (*bottom*) line traces of the conductance map in **b** taken at $B = 3$ T (*green*), $B = 4.5$ T (*pink*) and $B = 5.5$ T (*brown*). As the helical gap is independent of disorder or interference effects, these and other anomalous conductance features average out in a 2D colorplot improving the visibility of the helical gap in **b** compared to the individual traces in **d**

QPC channel, which sets the subband occupation, is controlled by applying a voltage to the gate $V_g$. The electric field in the nanowire, $E$, generated by the backgate and the substrate that the nanowire lies on, both induce a structural inversion asymmetry that results in a finite Rashba spin-orbit field. As the wire is translationally invariant along its length, the spin-orbit field, $B_{SO}$, is perpendicular to both the electric field and the wire axis. The effective channel length, $L_{QPC} \sim 245$ nm, as well as the shape of the onset potential $\lambda \sim 80$ nm, are set by electrostatics which are influenced by both the thickness of the dielectric and the amount of electric field screening provided by the metallic contacts to the nanowire (Fig. 1b). Here, we report measurements from one device. Data from an additional device that shows the same effect, as well as control devices of different channel lengths and onset potentials, is provided in the Supplementary Figs. 5–7.

The energy-momentum diagrams in Fig. 1c–e show the dispersion from the one-dimensional (1D) nanowire model of refs. [4, 5] including both SOI with strength $\alpha$ and Zeeman splitting $E_Z = g\mu_B B$, where $g$ is the g-factor, $\mu_B$ the Bohr magneton and $B$ the magnetic field strength. These dispersion relations explain how the helical gap can be detected: without magnetic field, the SOI causes the first two spin degenerate subbands to be shifted laterally in momentum space by $\pm k_{SO} = m^*\alpha/\hbar^2$ with $m^*$ the effective electron mass, as electrons with opposite spins carry opposite momentum, as shown in Fig. 1c. The corresponding spin-orbit energy is given by $E_{SO} = \hbar^2 k_{SO}^2/2m^*$. However, here Kramers degeneracy is preserved and hence the plateaus in

conductance occur at integer values of $G_0 = 2 e^2/h$, as for a system without SOI. Applying a magnetic field perpendicular to $B_{SO}$ the spin bands hybridize and a helical gap, of size $E_Z$ opens as shown in Fig. 1d. When the chemical potential $\mu$ is tuned by the external gate voltage, it first aligns with the bottom of both bands resulting in conductance at $1 \cdot G_0$ before reducing from $1 \cdot G_0$ to $0.5 \cdot G_0$ when $\mu$ is positioned inside the gap. This conductance reduction with a width scaling linearly with increasing Zeeman energy, is a hallmark of transport through a helical state. When the magnetic field is orientated at an angle $\theta$ to $B_{SO}$, the size of the helical gap decreases as it is governed by the component of the magnetic field perpendicular to $B_{SO}$, as shown in Fig. 1e. Additionally, the two sub-band bottoms also experience a spin splitting giving rise to an additional Zeeman gap. For a general angle $\theta$, the QPC conductance thus first rises from 0 to $0.5 \cdot G_0$, then to $1 \cdot G_0$, before dropping to $0.5 \cdot G_0$ again and the helical gap takes the form of a re-entrant $0.5 \cdot G_0$ conductance feature. By comparing to a 1D nanowire model, we can extract both the size of the helical gap $E_{helical} \approx E_Z \sin\theta$ and the Zeeman shift $E_{Zeeman} \approx E_Z \cos\theta$ (Supplementary Note 2 and Supplementary Figs. 1, 9 and 10). This angle dependency is a unique feature of SOI and can be used as a decisive test for its presence in the experimental data.

**Magnetic field dependence of the helical gap.** Figure 2 shows the differential conductance d$I$/d$V$ of our device at zero source-drain bias as a function of gate and magnetic field. Here and in the data

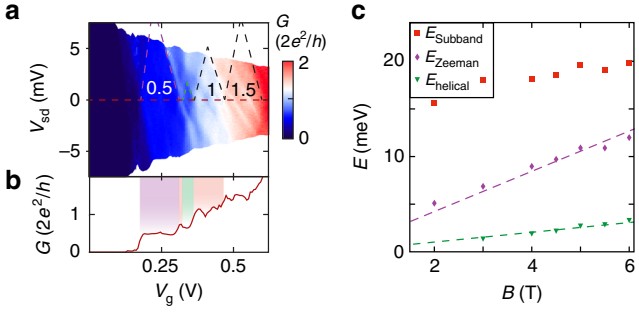

**Fig. 3** Voltage bias spectroscopy of the helical gap. **a** Conductance measurement as a function of QPC gate $V_g$ and source-drain bias voltage $V_{sd}$ at $B = 4$ T. The observed helical gap is a stable feature in voltage bias. *Dotted lines* are drawn as guide to the eye indicating the linecut presented in **b** (*red*), as well as the edges of the $0.5 \cdot G_0$ plateau (*purple*), helical gap (*green*) and the $1 \cdot G_0$ and $1.5 \cdot G_0$ plateaus (*black*). **b**, Linecut of **a** taken at $V_{sd} = 0$ mV together with the color scheme explained in Fig. 1. **c** Evolution of the energy levels extracted from the scans similar to **a**, at increasing magnetic field. Fits with intercept fixed at zero (*dotted lines*) give the *g*-factor of the first subband and the offset angle via $g = 1/(\mu_B \cos\theta) \cdot dE/dB$ and $E_{helical}/E_{Zeeman} \approx \tan\theta$. We find $g = 38 \pm 1$ and $\theta = 13° \pm 2°$. Individual scans are shown in Supplementary Fig. 2

shown in Fig. 3, the magnetic field B is offset at a small angle $\theta = 17°$ from $\boldsymbol{B}_{SO}$ in the $x$–$y$ plane (Fig. 2a). We determine that our device has this orientation from the angle dependence of the magnetic field, by clearly resolving the $1 \cdot G_0$ plateau before the re-entrant conductance feature, which is reduced at larger angles (Fig. 4 and Supplementary Note 2 and Supplementary Fig. 10). For low magnetic fields, we observe conductance plateaus quantized in steps of $0.5 \cdot G_0$, as typical for a QPC in a spin polarizing B-field with or without SOI. Above $B = 3$ T, the $1 \cdot G_0$ plateau shows a conductance dip to $0.5 \cdot G_0$. This re-entrant conductance feature evolves continuously as a function of magnetic field, before fully enveloping the $0.5 \cdot G_0$ plateau for magnetic fields larger than around 5.5 T. Line traces corresponding to the colored arrows in Fig. 2b are shown in Fig. 2d. The feature is robust at higher temperatures up to 1 K, as well across multiple thermal cycles (Supplementary Fig. 3).

Using the 1D nanowire model with $\theta = 17°$, we find that the helical gap feature vanishes into a continuous $0.5 \cdot G_0$ plateau when $E_Z = 2.4 E_{SO}$. Using the extracted g-factor $g = 38$ of our device (Fig. 3 and Supplementary Note 2), we find a lower bound for the spin-orbit energy $E_{SO} = 5.5$ meV, corresponding to a spin-orbit length $l_{SO} = 1/k_{SO} \approx 22$ nm. For a second device, we extract a similar value $E_{SO} = 5.2$ meV (Supplementary Figs. 5 and 6). Recently, it has been highlighted that the visibility of the helical gap feature depends crucially on the shape of the QPC potential[16]. To verify that our observation is compatible with SOI in this respect, we perform self-consistent simulations of the Poisson equation in Thomas–Fermi approximation for our device geometry. The resulting electrostatic potential is then mapped to an effective 1D QPC potential for a quantum transport simulation using parameters for InSb (for details, see Supplementary Note 1 and Supplementary Fig. 1). These numerical simulations, shown in Fig. 2c, fit best for $l_{SO} = 20$ nm ($E_{SO} = 6.5$ meV) and agree well with the experimental observation, corroborating our interpretation of the re-entrant conductance feature as the helical gap.

Voltage bias spectroscopy, as shown in Fig. 4a confirms that this state evolves as a constant energy feature. By analyzing the size of conductance triangles in voltage bias spectroscopy data at a range of magnetic fields, we directly convert the development of the initial $0.5 \cdot G_0$ plateau, as well as the re-entrant conductance feature to energy (Fig. 4c). From the evolution of the width of the

first $0.5 \cdot G_0$ plateau, we can calculate the g-factor of the first subband $g = 38 \pm 1$. This number is consistent with the recent experiments, which reported g-factors of 35–50[19, 20]. Comparing the slopes of the Zeeman gap and the helical gap $E_h/E_Z \approx \tan\theta$ provides an alternative way to determine the offset angle $\theta$. We find $\theta = 13° \pm 2°$ which is in reasonable agreement with the angle determined by magnetic field rotation.

**Angle dependence of the helical gap**. To confirm that the re-entrant conductance feature agrees with spin-orbit theory, we rotate the magnetic field in the plane of the substrate at a constant magnitude $B = 3.3$ T, as shown in Figs. 4a, b. When the field is rotated towards being parallel to $\boldsymbol{B}_{SO}$, the conductance dip closes, while when it is rotated away from $\boldsymbol{B}_{SO}$, the dip increases in width and depth. In contrast, when the magnetic field is rotated the same amount around the $y$–$z$ plane, which is largely perpendicular to $\boldsymbol{B}_{SO}$, there is little change in the re-entrant conductance feature, as shown in Figs. 4c, d. Rotating through a larger angle in the $x$–$y$ plane (Figs. 4e, f) shows that this feature clearly evolves with what is expected for spin orbit. Our numerical simulations in Fig. 4g agree well with the observed experimental data. The small difference in the angle evolution between the numerical simulations and experimental data can be attributed to imperfect alignment of the substrate with the $x$–$y$ plane.

**Discussion**

The extracted SO energy of 6.5 meV is significantly larger than that obtained via other techniques, such as weak anti-localization (WAL) measurements[21], and quantum dot spectroscopy[20]. This is not entirely unexpected, due to the differing geometry for this device and different conductance regime it is operated in. Quantum dot measurements require strong confinement, and so the Rashba SOI is modified by the local electrostatic gates used to define the quantum dot. Weak anti-localization measurements are performed in an open conductance regime, however they assume transport through a diffusive, rather than a ballistic channel. Neither of these measurements explicitly probe the spin-orbit interaction where exactly one mode is transmitting in the nanowire, the ideal regime for Majoranas, and so the spin-orbit parameters extracted from QPC measurements offer a more accurate measurement of the SOI experienced by the Majorana zero mode. Also, the SOI in a nanowire can be different for every subband, and it is expected that the lowest mode has the strongest SOI due to a smaller confinement energy[3]. Additionally, the finite diameter of the nanowire, together with impurities within the InSb crystal lattice[23] both break the internal symmetry of the crystal lattice which can modify the SOI and may contribute a non-zero Dresselhaus component to the spin-orbit energy that has not been previously considered.

While high quality quantized conductance measurements have been previously achieved in short channel devices[17] ($L \sim 150$ nm), the channel lengths required for observing the helical gap are at the experimental limit of observable conductance quantization. As shown in Supplementary Figs. 1, 8 and 10, small changes in the QPC channel length, spin-orbit strength or the QPC potential profile are enough to obscure the helical gap, particularly for wires with weaker SOI. We have fabricated and measured a range of QPCs with different length and potential profiles, and only two devices of $L \sim 300$ nm showed unambiguous signatures of a helical gap. Possibly some of the other devices did not show clear signatures because they had weaker SOI.

Several phenomena have been reported to result in anomalous conductance features in a device such as this. Oscillations in conductance due to Fabry–Perot resonances are a common feature in clean QPCs. Typically the first oscillation at the front of

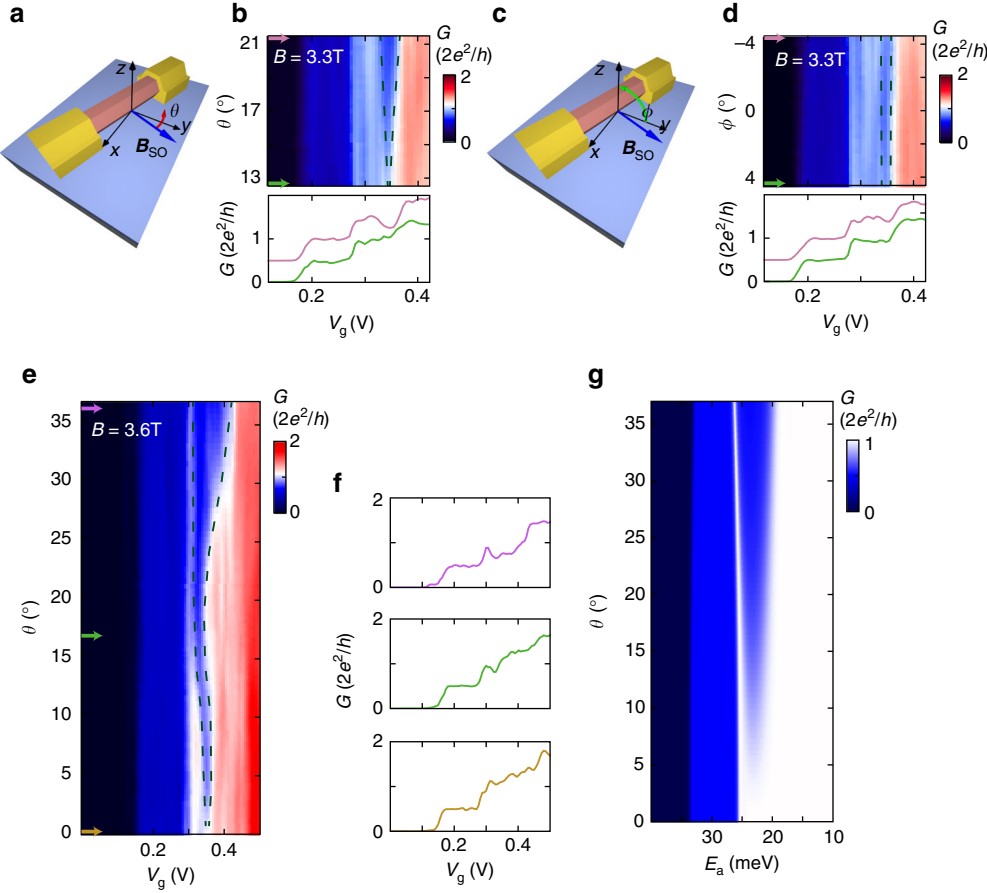

**Fig. 4** Angle dependence of the helical gap. **a** Illustration of the magnetic field orientation in **b**. When rotating along $\theta$ the magnetic field lies in the $x$–$y$ plane and is aligned with $\boldsymbol{B}_{SO}$ at $\theta = 0°$ **b**, Rotation of the magnetic along $\theta$ at $B = 3.3$ T, shows a strong angle dependence of the helical gap. The conductance dip closes when $\boldsymbol{B}$ is rotated toward $\boldsymbol{B}_{SO}$ and opens when $\boldsymbol{B}$ is rotated away from $\boldsymbol{B}_{SO}$. Linecuts taken at $\theta = 12.9°$ (*green arrow*) and $\theta = 21.1°$ (*pink arrow*) are shown in the *bottom panel*. **c** Illustration of the magnetic field orientation in **d**. When rotating along $\Phi$ the magnetic field lies in the $x$–$z$ plane. At $\Phi = 0°$ $\boldsymbol{B}$ is aligned with the $y$-axis at at a small offset angle relative to $\boldsymbol{B}_{SO}$. **d** Rotation of the magnetic field along $\Phi$ at $B = 3.3$ T, mostly perpendicular to $\boldsymbol{B}_{SO}$, with linecuts at $\Phi = -4.1°$ (*pink arrow*), and $\Phi = 4.1°$ (*green arrow*), added in the *bottom panel*. While the angle range is identical to **b**, there is little change in the conductance dip. **e** Rotation of the magnetic field along $\theta$ at $B = 3.6$ T over a large angle range. The conductance dip disappears when $\boldsymbol{B}$ is parallel to $\boldsymbol{B}_{SO}$ which gives the exact offset angle between $\boldsymbol{B}_{SO}$ and $\boldsymbol{B}_Z$, $\theta = 17°$. **f** Linecuts of **e** taken at $\theta = 0°$ (*brown arrow*), $\theta = 17°$ (*green arrow*), and $\theta = 37°$ (*pink arrow*), additional linecuts are shown in Supplementary Fig. 4. **g** Numerical simulations of the differential conductance in a magnetic field rotated along $\theta$ with $L = 325$ nm and $l_{SO} = 20$ nm. *Black dashed lines* indicating the width of the helical gap are added as guide to the eye in **b**, **d**, **e**

each plateau is the strongest and the oscillations monotonically decrease in strength further along each plateau[16, 22]. In our second device, we clearly observe Fabry–Perot conductance oscillations at the beginning of each plateau, however these oscillations are significantly weaker than the subsequent conductance dip. Furthermore we observe Fabry–Perot oscillations at each conductance plateau, while the re-entrant conductance feature is only present at the $1 \cdot G_0$ plateau. Additionally, the width of the Fabry–Perot oscillations does not change with increasing magnetic field, unlike the observed re-entrant conductance feature. A local quantum dot in the Coulomb or Kondo regimes can lead to conductance suppression, which increases in magnetic field[24]. However both effects should be stronger in the lower conductance region, and exists at zero magnetic field, unlike the feature in our data. Additionally, a Kondo resonance should scale with bias voltage $V_{sd} = \pm g\mu_B B/e$ as a function of external magnetic field, decreasing instead of increasing the width of the region of suppressed conductance. Given the $g$-factor measured in InSb quantum dots, and its variation with the angle of applied magnetic field $g = 35$–$50$[20], we can exclude both these effects. Similarly the Fano effect and disorder can also induce a conductance dip, but these effects should

not increase linearly with magnetic field. The 0.7 anomaly occurs at the beginning of the plateau, and numerical studies have shown it does not drastically affect the observation of the helical gap[25].

In conclusion, we have observed a return to $1\,e^2/h$ conductance at the $2\,e^2/h$ plateau in a QPC in an InSb nanowire. The continuous evolution in increasing magnetic field and the strong angle dependence in magnetic field rotations agree with a SOI related origin of this feature and distinguish it from Fabry–Perot oscillations and other $g$-factor related phenomena. Additional confirmation is given by numerical simulations of an emerging helical gap in InSb nanowires. The extracted spin-orbit energy of 6.5 meV is significantly larger than what has been found by other techniques, and more accurately represents the true spin-orbit energy in the first conduction mode. Such a large spin-orbit energy reduces the requirements on nanowire disorder for reaching the topological regime[26], and offers promise for using InSb nanowires for the creation of topologically protected quantum computing devices.

## Methods
**Device fabrication**. The InSb nanowires were grown using the metalorganic vapor phase epitaxy (MOVPE) process, and are grown along the [111] growth direction in a zincblende crystal structure[27]. The InSb nanowires were deposited

using a deterministic deposition method on a degenerately doped silicon wafer. The wafer covered with 20 nm of low stress LPCVD SiN which is used as a high quality dielectric. Electrical contacts (Cr/Au, 10 nm/110 nm) defined using ebeam lithography were then evaporated at the ends of the wire. Before evaporation the wire was exposed to an ammonium polysulfide surface treatment and short helium ion etch to remove the surface oxide and to dope the nanowire underneath the contacts[17].

**Measurements**. Measurements are performed in a dilution refrigerator with base temperature ~20 mK fitted with a 3-axis vector magnet, which allowed for the external magnetic field to be rotated in-situ. The sample is mounted with the substrate in the $x$–$y$ plane with the wire orientated at a small offset angle $\theta = 17°$ from the $x$-axis. We measure the differential conductance $G = dI/dV$ using standard lock-in techniques with an excitation voltage of 60 µV and frequency $f = 83$ Hz. Additional resistances due to filtering are subtracted to give the true conductance through the device. The helical gap, Zeeman gap, and subband spacing reported in Fig. 3c were extracted from analysis of the full voltage bias conductance diamonds shown in Fig. 3a and Supplementary Fig. 2. The subband spacing was extracted by summing the widths of the 0.5 and 1 plateaus, the helical gap and Zeeman gap from their respective conductance diamonds.

**Numerical transport simulations**. We use the method of finite differences to discretize the 1D nanowire model of ref. [5]. In order to obtain a 1D QPC potential, we solve the Poisson equation self-consistently for the full three-dimensional (3D) device structure treating the charge density in the nanowire in Thomas–Fermi approximation. To this end, we use a finite element method, using the software FEniCS[28]. The resulting 3D potential is then projected onto the lowest nanowire subband and interpolated using the QPC potential model of ref. [16]. Transport in the resulting tight-binding model is calculated using the software Kwant[29].

**Code availability**. All code used for the simulations in this study is available from the 4TU.ResearchData repository at doi:10.4121/uuid:f82b6a24-201f-4de7-94cb-afc95ad1adea (http://doi.org/10.4121/uuid:f82b6a24-201f-4de7-94cb-afc95ad1adea).

**Data availability**. All data underlying this study are available from the 4TU.ResearchData repository at doi:10.4121/uuid:686925fd-017c-49df-a92b-3dc84138c513 (http://doi.org/10.4121/uuid:686925fd-017c-49df-a92b-3dc84138c513).

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

## Acknowledgements

We gratefully acknowledge D. Xu, S. Goswami, D. van Woerkom and R.N. Schouten for their technical assistance and helpful discussions. This work has been supported by funding from the Netherlands Foundation for Fundamental Research on Matter (FOM), the Netherlands Organization for Scientific Research (NWO/OCW), the Office of Naval Research, Microsoft Corporation Station Q, the European Research Council (ERC) and an EU Marie-Curie ITN.

## Author contributions

J.K. and F.P. fabricated the samples, J.K., M.C.C., and F.P. performed the measurements with input from M.W. Ö.G. contributed to the experiments. M.W., M.N., and A.V. developed the theoretical model and performed simulations. D.C., S.R.P., and E.P.A.M.B. grew the InSb nanowires. All authors discussed the data and contributed to the manuscript.

## Additional information

**Competing interests:** The authors declare no competing financial interests.

