## [Peer Review File · Nature Communications]

Reviewers' comments:

Reviewer #1 (Remarks to the Author):

The authors present a paper, nicely showing and explaining different conductance plateaus in InSb nanowire. I think these results are very important for the progress in realizing a possible topological quantum computer with Majorana modes, and thus recommend the publication in Nature Communications. My only confusion is about two distinct values of the angle θ , reported in the paper - can the authors maybe be more explicit about why they claim 13 and 17 are almost the same?

Reviewer #2 (Remarks to the Author):

Authors present experimental results on very important and timely topic: helical conductors. They are important ingredient for Majorana fermions, and so far have not yet been properly demonstrated in experiment. If such a demonstration were presented, it would indeed be of great and broad interest, and break-through result, to publish without delay. However, many open questions and inconsistencies are in this manuscript, detailed below (not in order of importance).

1. Fig S3a: why are Zeeman lines closing at $B \sim 2T$ and NOT at $B=0$?
2. no Zeeman gap at $B=3T$ in data, but finite Zeeman in numerics? This seems inconsistent.
3. Conductance in experiment is above $2e^2/h$ right around/after the 'helical reentrance' at larger V_g , while theory remains at $\leq 2e^2/h$. The theory is a single subband model. Does this difference between exp. and th means that the experiment is not in single subband regime? Only more than one subband can lead to conductance above $2e^2/h$. Color codes of experiment, eg Fig2 and 4, goes up to $4 e^2/h$ but in the fit-model only to $2e^2/h$.
4. Could the so called "helical reentrance" just be a random peak, for example due to Fabri-Perrot or disorder, and/or mixing in from higher bands?
5. Fig. 3B: subband energy should move DOWN in B field? (opposite slope to the lowest subband)
6. Fig with energy distance between subbands is misleading: the distance becomes smaller when gate voltage value is taken where conductance exceeds $2e^2/h$ (rather than value far inside plateau).
7. Arbitrary direction of applied field: why 17 degrees? Should be 90 degrees (optimal angle, also used in Majorana experiment)
8. Reentrance feature is still clearly present at 0 degrees (dip in colorscale conductance Fig 4c). This is inconsistent with helical gap model, see theory 4d, there is no dip at 0 degrees. The dashed lines (added by hand) are suggestive and deceptive.
9. Rather than showing angle $0 \leq \theta \leq 37$ degrees, a broad range up to 90 degrees should be investigated, as the effect most pronounced at 90.
10. The rotation of B field leads to effects related to g factor anisotropy: the g factor in different direction is different. Was this checked, and how?
11. Such models are typically sensitive to the effect of disorder, Fabry_perot, temperature, and gate shape, etc Were such effects investigated in the modeling? Can these be separated from "reentrance" feature?
12. A gigantic SO of 6 meV is found. How is this possible? The wire is Wurtzite, and thus has no intrinsic Rashba or Dresselhaus SOI, see Fasth, Fuhrer PRL (also Wurtzite). So SO is induced by some electric fields, but here there are no local gates, only a back gate quite far away, yet a huge SOI is claimed.

Publication cannot be recommended until these serious issues can be resolved.

Reviewer #3 (Remarks to the Author):

The manuscript describes experimental transport measurements on an InSb nanowire, part of which functions as a quantum point contact, and shows that for certain combinations of gate voltage and magnetic field, transport is compatible with conductance through a helical state. The experiments are supported by model calculations and simulations (largely detailed in the supplementary materials). The work presents new data of high current interest and is convincing, and is a good fit for Nature Communications. The experimental approach is based on known techniques for quantum point contacts, here used to explore the helical state. The value lies in a convincing demonstration of spin-polarized transport in a low-dimensional constriction via the helical state, which has been described in the prior literature, especially theoretically so far. Spin-polarized transport under spin-orbit coupling has given rise to some confusion also in the prior literature, and the present work helps clear the confusion. The work also helps stating fairly clear conditions for the observation of the helical state. For instance the authors mention the limited number of devices showing the helical state among a much larger number they tried, and explain which parameters play a role. They also present data on an additional device, differing in important details, that likewise shows the helical state. Further afield, the manuscript establishes valuable data for the exploration of new quantum states of matter in spin-orbit coupled nanowires (such as Majorana states), and demonstrates a use of high-quality nanowires. The experiments and supporting theoretical aspects are quite complete and sufficiently circumspect, and convincingly eliminate other phenomena with similar features that could confuse the analysis of the re-entrant $\frac{1}{2} G_0$ observation. The following are minor observations that will make the manuscript easier to understand:

- 1) It will be useful to briefly yet explicitly state in the main text how the calibration is performed from gate voltage to energy (sometimes called "voltage lever"). And, if applicable, to state that the conversion from drain-source voltage to energy is direct. More elaboration can be given in Suppl. Mat. Since the conversion is quite central to the data interpretation in terms of energy states in a Hamiltonian, it deserves more clear and explicit explanation. Figure 4 hints at the conversion, and Suppl. Mat. p. 4 (top) mentions the usual QPC diamonds, and Fig. S5 contains calibration data for Fig. S4. But the information is dispersed and not sufficiently explicit, and not mentioned in the main text.
- 2) In the main text it will be useful to state explicitly that for Fig. 2, B is applied in the x-y plane, referring to Fig. 1a. It is only after reading the Suppl. Mat. (e.g. Eq. S5), or after seeing Fig. 4, that this becomes more unambiguous.
- 3) In Suppl. Mat., the authors pay attention to the potential along x. But what about the potential in the y-z plane, so normal to the wire? Specifically, what is the depletion layer width at the wire boundaries? Is it of the same order as λ ? Is the remaining effectively conducting width or diameter of the wire of the order of $\frac{1}{2}$ the Fermi wavelength, so that it is compatible with QPC operation?
- 4) The energy band picture inset in Fig. 1b is not explained and not used. It is potentially useful however. The chemical potential μ has the same symbol as the magnetic moment due to a typo in the latter (*italic* on line 3 of the manuscript, *straight* on line 4).
- 5) Typo in "Strength", Suppl. Mat. contents table, VII.

Reviewer #1:

The authors present a paper, nicely showing and explaining different conductance plateaus in InSb nanowire. I think these results are very important for the progress in realizing a possible topological quantum computer with Majorana modes, and thus recommend the publication in Nature Communications. My only confusion is about two distinct values of the angle theta, reported in the paper - can the authors maybe be more explicit about why they claim 13 and 17 are almost the same?

We thank the Reviewer for the time invested in reading and commenting on our paper and their recommendation that it be published in Nature Communications.

The angle theta is extracted in two different ways from the experimental data. The first estimate of 17 degrees is made from the magnetic field rotation data in Fig 4, where we take the angle for which the helical gap is smallest to be 'aligned' with B_{SO} . This is at ~ 17 degrees relative to the magnetic field sweep in Fig 2. There is some error in this estimation which is difficult to quantify, but we would estimate it visually to be $\sim \pm 2$ degrees. For example, if the substrate that the nanowire lies on is not aligned with the x-y plane, the helical gap may never close completely. We did not include this error estimate in the original manuscript as it was not determined in a quantifiable way. The second estimation of the angle of 13 ± 2 degrees comes from comparing the evolution of the Zeeman vs helical gaps in Fig 3b. This is an independent method of extracting the sample orientation. Given the errors associated with each extraction technique, we believe it is justified to say that the two different angles are comparable.

Reviewer 2:

Authors present experimental results on very important and timely topic: helical conductors. They are important ingredient for Majorana fermions, and so far have not yet been properly demonstrated in experiment. If such a demonstration were presented, it would indeed be of great and broad interest, and break-through result, to publish without delay. However, many open questions and inconsistencies are in this manuscript, detailed below (not in order of importance).

We thank the reviewer for the detailed comments on our paper and are pleased that they agree that our results are on a very important and timely topic. We agree with the reviewer that a careful analysis of the data is important to exclude alternative explanations of our measurements.

Below we provide further clarification and discussion for these alternative scenarios that we hope will satisfy the reviewer's concerns. We believe that we are able to convincingly exclude these alternative explanations for our measurements, and so we remain confident in our claim of the observation of a helical gap.

1. Fig S3a: why are Zeeman lines closing at $B \sim 2T$ and NOT at $B=0$?

The reviewer correctly points out that the Zeeman gap should start around $B = 0T$ in an ideal, infinite QPC. However, as shown in Rainis et al. (Ref 18) and our simulations, a finite QPC length will cause deviation from this ideal case. The simulations in Fig S8 and S10 of the Supplementary Information of our paper show that both the QPC length and the strength of the SOI strongly influence the magnetic field strength at which the helical gap becomes visible. In short QPCs or for weak SOI, the helical gap size is too small to suppress electron tunnelling and it remains invisible at low magnetic fields. The lines in Fig S3a were added as guide to the eye and follow the experimental visibility of the gap. Extracting the size of the helical gap from voltage bias spectroscopy measurements as presented in Fig 3b and S5c, is more reliable and confirms that the gap evolves linearly with an extrapolated closing at $B = 0T$.

2. no Zeeman gap at $B=3T$ in data, but finite Zeeman in numerics? This seems inconsistent.

The simulations in Fig 2 and 4 of the main text of our manuscript present results from an ideal defect free model. In a realistic experimental system, local defects or scattering centers can interrupt or change the length of the gapped region. This effect is enhanced at lower magnetic field where the Zeeman gap is smaller. This backscattering reduces or even destroys the experimental gap visibility, similar to the case of too short QPCs (Fig. S8). At higher magnetic field, backscattering effects become small compared to the Zeeman gap and the visibility is therefore closer to the ideal case. Since the visibility of the helical gap depends strongly on details of the gate potential, small experimental deviations from the ideal potential further contribute to differences between experiment and simulations.

3. Conductance in experiment is above $2e^2/h$ right around/after the 'helical reentrance' at larger V_g , while theory remains at $\leq 2e^2/h$. The theory is a single subband model. Does this difference between

exp. and th means that the experiment is not in single subband regime? Only more than one subband can lead to conductance above $2e^2/h$. Color codes of experiment, eg Fig2 and 4, goes up to $4 e^2/h$ but in the fit-model only to $2e^2/h$.

We thank the reviewer for this comment. The numerical model presented in our paper only includes the first sub-band of the QPC, which is most relevant for understanding our results. Although possible, including higher sub-bands in the model adds significantly to the calculation time without adding to the interpretation of the results as the presence of the helical gap of the first sub-band is unaffected by transport in higher sub-bands. The experimental data from the nanowire devices does, of course, include these higher sub-bands. As an example, we note that at higher gate voltages we do see a 1.5 plateau in the experimental data, which was not included in the single sub-band model.

Because we measure long QPCs, resonances and backscattering may cause these higher sub-bands to not be clearly resolved at the exact quantization values expected (see Rainis et al. for numerical studies on the effects of disorder on nanowire QPCs). To additionally confirm our interpretation, we also include voltage spectroscopy measurements at varying magnetic field. The increased averaging at high bias voltages reduces such deviations and as shown in Fig S2 and S6 and we clearly observe diamond shaped plateaus of constant conductance together with a diamond shaped helical re-entrance feature.

4. Could the so called "helical reentrance" just be a random peak, for example due to Fabri-Perrot or disorder, and/or mixing in from higher bands?

We thank the reviewer for bringing up this very important point.

In our manuscript we provide the following data that clearly distinguish the evolution of a helical gap from other effects such as Fabry-Perot oscillations or disorder.

1. In Figure 2, the feature evolves linearly with magnetic field, which is consistent with the explanation of a helical gap. Disorder effects or Fabry-Perot oscillations should stay constant in magnetic field, or should disappear at increasing fields due to reduced effects of scattering.
2. In Figure 3, the feature evolves correctly with respect to the g-factor and the orientation of the nanowire, and shows up clearly as a diamond shaped region of constant conductance in voltage bias spectroscopy. This would not occur if the feature was due to disorder
3. In Figure 4, the feature evolves correctly with the rotation of the magnetic field. Disorder effects or Fabry-Perot would not have such an angular magnetic field dependence, neither would an effect of mixing in from higher bands.

The combination of all these features and their robustness after thermal cycles, together with their reproducibility in a second device, make us confident that we have in fact observed a helical re-entrance feature. A disorder feature which fulfils all these conditions is, in our opinion, highly unlikely.

5. Fig. 3B: subband energy should move DOWN in B field? (opposite slope to the lowest subband)

We thank the reviewer for this comment and apologize if Fig 3b was not sufficiently explained. The sub-band energy drawn in red in Fig 3b (and Fig S5), was calculated by summing the energy of the 0.5 and the 1 plateau. These values were obtained from the respective diamond sizes in the scans shown in Fig S2 (S6). Earlier experiments on nanowire QPCs (e.g. Van Weperen et al, Ref. 21) have shown that this energy spacing stays mostly constant with increasing magnetic field.

6. Fig with energy distance between subbands is misleading: the distance becomes smaller when gate voltage value is taken where conductance exceeds $2e^2/h$ (rather than value far inside plateau).

Again we apologize if Fig 3b was not sufficiently explained. The sub-band energy values were not extracted from the individual low-bias conductance traces, but from full bias scans, as shown in Fig S2 (S6). By measuring the size of the conductance diamonds at higher bias voltage, we decrease the influence of resonances and conductance fluctuations. This provides a more reliable method for extracting the sub-band spacing, and is used commonly throughout the literature. The fact that both fits in Figures 3 and S5 pass through the origin and agree well with the data additionally confirms our extracted values.

To clarify this we have provided the following explanation in the main text

“By analysing the size of conductance triangles in voltage bias spectroscopy data at a range of magnetic fields, we directly convert the development of the initial $0.5 \cdot G_0$ plateau, as well as the reentrant conductance feature to energy (Fig 3b).”

and in the Methods section of the manuscript.

“The helical gap, Zeeman gap, and subband spacings reported in Fig 3b were extracted from analysis of the full voltage bias conductance diamonds shown in Fig 3a and Fig S2. The subband spacing was extracted by summing the widths of the 0.5 and 1 plateaus, the helical gap and Zeeman gap from their respective conductance diamonds.”

7. Arbitrary direction of applied field: why 17 degrees? Should be 90 degrees (optimal angle, also used in Majorana experiment)

The alignment angle was not chosen at random, but rather optimized for the specific experimental conditions of the helical gap measurements, which are different from the Majorana experiments. As seen in our numerical simulations in Fig S9, when the magnetic field oriented at 90 degrees, the onset of the helical gap occurs at, or close to, zero magnetic field. Because our measurements require long QPCs, near the experimental limit, the conductance plateaus at low magnetic field are strongly affected by disorder, whereas conductance plateaus at higher magnetic fields are clearer to identify. Therefore, by aligning the nanowire at a small angle (~ 10 - 20 degrees), the helical gap will appear at a slightly higher magnetic field value in a region where disorder and scattering are suppressed and clear plateaus can be identified. As our experimental setup limits the accessible angle range over which the magnetic field can be rotated the optimal alignment is chosen close to the strongest angle response of the helical gap. This is around 0 degrees and not 90 degrees.

These two factors made us aim for a small misalignment of $\theta \sim 10 - 20$ degrees. The precise experimental alignment of 17 (11) degrees results from experimental uncertainties, such as the precision during wire deposition and while mounting the sample. This can be done with an accuracy of about 5 degrees.

8. Reentrance feature is still clearly present at 0 degrees (dip in colorscale conductance Fig 4c). This is inconsistent with helical gap model, see theory 4d, there is no dip at 0 degrees. The dashed lines (added by hand) are suggestive and deceptive.

The helical gap will only vanish when the external magnetic field is aligned perfectly with the spin orbit field, B_{SO} . In our experimental setup, we have a limited range over which we can rotate the magnetic field in situ, and so it is possible that we never reach perfect alignment. In this case a small conductance dip will remain, as seen in our data. This could be either because of a small remaining misalignment in the x-y plane as the vector magnet does not reach a large enough rotation angle, or because there is a small misalignment between the sample plane and the magnet's x-y plane. Nevertheless, the trend of the gap closing as the magnetic field is rotated is clearly present in the data, which supports our interpretation of this feature resulting from a spin-orbit phenomena such as the helical gap. The dashed lines in Fig 4c were drawn based on this strong angle dependence.

We have now added a clarification of this in Fig S4 of the supplementary information, together with additional linecuts of Fig 4c that emphasize the strong angle dependence around 0 degrees.

9. Rather than showing angle $0 \leq \theta \leq 37$ degrees, a broad range up to 90 degrees should be investigated, as the effect most pronounced at 90.

We agree with the reviewer that investigating such a large angle range would be preferable. Unfortunately, we do not possess a dilution refrigerator with a vector magnet able to rotate at 3.5 T in a full 3D sphere. We hope that our work prompts other experimentalists who may have such a system to perform these measurements in the future.

10. The rotation of B field leads to effects related to g factor anisotropy: the g factor in different direction is different. Was this checked, and how?

The existence and possible influence of a g-factor anisotropy in InSb nanowire QPC is an interesting but open question, that has to our knowledge so far not been studied experimentally. Existing reports of anisotropic g-factors in nanowires all investigate the g-factor of quantum dot states where strong anisotropic confinement influences the g-factor, for example leading to a change in g-factor of ~25% in InSb quantum dots (Nadj-Perge, PRL 2012).

In contrast to this, for the lowest sub-band, the confinement due to the finite nanowire diameter plays only a small role. A recent numerical study (Winkler arxiv 1703.10091) confirmed this and found only a minor g-factor anisotropy. For a wire with 40nm diameter the g-factor changed by less than 5% when rotating the magnetic field by 90 degrees. The wires used in our experiments have a diameter $d \sim 100$ nm which is significantly larger than 40nm, therefore we expect an even smaller anisotropy of the g-factor.

Experimentally one can study the evolution of the 0.5 plateau in a rotating magnetic field. A g-factor anisotropy would lead to a sizeable change in the width of this plateau, as seen in InSb 2DEGs (F.Qu Nano Letters 2016). The linecuts presented in Figure 4a,c show only small width changes which are mostly related to a steeper onset of the plateau at smaller angles. Importantly, in Fig 4b, when the magnetic field is rotated (mostly) perpendicular to the wire axis no anisotropy is expected and we still see a small change, comparable to the change in Fig 4a,c.

This experimental check in combination with the theoretical predictions make us confident in assuming that effects related to an anisotropic g-factor are small or negligible.

11. Such models are typically sensitive to the effect of disorder, Fabry_perot, temperature, and gate shape, etc. Were such effects investigated in the modeling? Can these be separated from "reentrance" feature?

Numerical modelling by Rainis et al (ref 18) thoroughly investigates the effects of disorder and Fabry-Perot oscillations. As shown by Rainis et al., these effects may conspire to limit the visibility of a helical gap feature. However, these effects result in somewhat random oscillations rather than a flat plateau at e^2/h , and are not expected to evolve linearly in an increasing magnetic field as we see in Fig 2 and 3, or show the specific angle dependence we see in Fig 4. Additionally we expect that the temperature dependence should follow the size of the energy gap. In Fig S3, we see that the helical gap feature remains at temperatures of 1K, as expected.

12. A gigantic SO of 6 meV is found. How is this possible? The wire is Wurtzite, and thus has no intrinsic Rashba or Dresselhaus SOI, see Fasth, Fuhrer PRL (also Wurtzite). So SO is induced by some electric fields, but here there are no local gates, only a back gate quite far away, yet a huge SOI is claimed.

We agree with the reviewer that this result is surprising, which is why we took the time to obtain measurements of a second device to confirm the result, as well as performing extensive numerical studies.

We would like to begin by clarifying some of the Reviewer's assumptions:

Firstly, the backgate is actually quite close, underneath only 20nm of SiN dielectric. Therefore, an extremely large electric field could be generated in the nanowire.

Secondly, the InSb nanowires used in this experiment are grown with zinc-blende crystal structure, not Wurtzite as suggested by the reviewer. We have now clarified this in the Methods section of the manuscript.

"The InSb nanowires were grown using the metalorganic vapor phase epitaxy (MOVPE) process, and are grown along the [111] growth direction in a zinc-blende crystal structure."

Although zinc-blende crystals are not expected to have a Dresselhaus SO interaction in the [111] growth direction, as we state in the introduction of the manuscript, there are several scenarios where this assumption breaks down. We address this in the discussion section of the manuscript with the following statement.

"Additionally, the finite diameter of the nanowire, together with impurities within the InSb crystal lattice both break the internal symmetry of the crystal lattice and may contribute a non-zero Dresselhaus component to the spin orbit energy that has not been previously considered."

Works such as those by Fasth, Fuhrer et al. as suggested by the Reviewer extract spin-orbit values from quantum dot measurements. Here the strong electrostatic confinement significantly reduces

the strength of the SOI. Because confinement is lowest in the first subband, its SOI is expected to be increased. This is already discussed in the concluding paragraphs of the manuscript:

“The extracted SO energy of 6.5 meV is significantly larger than that obtained via other techniques, such as weak anti localization (WAL) measurements,²³ and quantum dot spectroscopy.²² This is not entirely unexpected, due to the differing geometry for this device and different conductance regime it is operated in. Quantum dot measurements require strong confinement, and so the Rashba SOI is modified by the local electrostatic gates used to define the quantum dot. Weak anti-localization measurements are performed in an open conductance regime, however they assume transport through a diffusive, rather than a ballistic channel. Neither of these measurements explicitly probe the spin orbit interaction where exactly one mode is transmitting in the nanowire, the ideal regime for Majoranas, and so the spin orbit parameters extracted from QPC measurements offer a more accurate measurement of the SOI experienced by the Majorana zero mode. Also, the SOI in a nanowire can be different for every subband, and it is expected that the lowest mode has the strongest SOI due to a smaller confinement energy.¹⁰”

Very recent measurements of our nanowires with atom probe spectroscopy (Koelling, Nano Letters 2017), found a residual As content of up to 4% in similarly grown InSb nanowires. Such contaminations could further modify SOI. We have now added this result to our list of references (ref 25).

“Additionally, the finite diameter of the nanowire, together with impurities within the InSb crystal lattice²⁵ both break the internal symmetry of the crystal lattice which can modify the SOI and may contribute a non-zero Dresselhaus component to the spin orbit energy that has not been previously considered.”

Furthermore, one has to take into account the selectivity of our measurements. As the simulations in Fig S10 show, even under ideal conditions, only wires with strong SOI will show a helical signature in our measurements. It is possible that some of our measured devices had a weaker SOI and therefore did not show a helical gap. We have now explicitly added this point to the manuscript.

“We have fabricated and measured a range of QPCs with different length and potential profiles, and only two devices of $L \sim 300$ nm showed unambiguous signatures of a helical gap. Possibly some of the other devices did not show clear signatures because they had weaker SOI.”

WAL (weak anti localization) measurements performed in diffusive 1D nanowires found an $E_{SO} \sim 10$ times larger than in Quantum Dots. Because E_{SO} depends quadratically on the SOI strength α , it exaggerates such differences. After converting our results to α ($\alpha_{helical, InSb} \sim 2.6 \text{ eV\AA}$), they are more similar to the results obtained through WAL ($\alpha_{WAL, InSb} \sim 0.5 - 1 \text{ eV\AA}$). We believe that the increased SOI of the first sub-band, together with the selectivity of our measurements can explain the remaining difference to earlier experiments. Finally we note that another experimental work studying the helical gap in InAs nanowires has recently appeared (Heedt et al, <https://arxiv.org/abs/1701.08439>, and online at Nature Physics doi:10.1038/nphys4070). Here the authors find an even larger difference between the values obtained by WAL and by the helical gap ($\alpha_{WAL, InAs} = 0.24 \text{ eV\AA}$; $\alpha_{helical, InAs} = 1.2 \text{ eV\AA}$) and they also find a surprisingly large SOI with $E_{SO} = 2.4 \text{ meV}$.

Reviewer #3:

The manuscript describes experimental transport measurements on an InSb nanowire, part of which functions as a quantum point contact, and shows that for certain combinations of gate voltage and magnetic field, transport is compatible with conductance through a helical state. The experiments are supported by model calculations and simulations (largely detailed in the supplementary materials). The work presents new data of high current interest and is convincing, and is a good fit for Nature Communications. The experimental approach is based on known techniques for quantum point contacts, here used to explore the helical state. The value lies in a convincing demonstration of spin-polarized transport in a low-dimensional constriction via the helical state, which has been described in the prior literature, especially theoretically so far. Spin-polarized transport under spin-orbit coupling has given rise to some confusion also in the prior literature, and the present work helps

clear the confusion. The work also helps stating fairly clear conditions for the observation of the helical state. For instance the authors mention the limited number of devices showing the helical state among a much larger number they tried, and explain which parameters play a role. They also present data on an additional device, differing in important details, that likewise shows the helical state. Further afield, the manuscript establishes valuable data for the exploration of new quantum states of matter in spin-orbit coupled nanowires (such as Majorana states), and demonstrates a use of high-quality nanowires. The experiments and supporting theoretical aspects are quite complete and sufficiently circumspect, and convincingly eliminate other phenomena with similar features that could confuse the analysis of the re-entrant $\frac{1}{2} G_0$ observation. The following are minor observations that will make the manuscript easier to understand:

We thank the Reviewer for their positive remarks and detailed reading of our manuscript and their recommendation that it be published in Nature Communications. Below we address each of their comments

1) It will be useful to briefly yet explicitly state in the main text how the calibration is performed from gate voltage to energy (sometimes called "voltage lever"). And, if applicable, to state that the conversion from drain-source voltage to energy is direct. More elaboration can be given in Suppl. Mat. Since the conversion is quite central to the data interpretation in terms of energy states in a Hamiltonian, it deserves more clear and explicit explanation. Figure 4 hints at the conversion, and Suppl. Mat. p. 4 (top) mentions the usual QPC diamonds, and Fig. S5 contains calibration data for Fig. S4. But the information is dispersed and not sufficiently explicit, and not mentioned in the main text.

We thank the reviewer for this helpful comment and agree that the conversion is central to the interpretation of our data. The corresponding description of Fig 3 in the main text has been adapted to explicitly include this and now reads:

“By analysing the size of conductance triangles in voltage bias spectroscopy data at a range of magnetic fields, we directly convert the development of the initial $0.5 \cdot G_0$ plateau, as well as the reentrant conductance feature to energy (Fig 3b).”

2) *In the main text it will be useful to state explicitly that for Fig. 2, B is applied in the x-y plane, referring to Fig. 1a. It is only after reading the Suppl. Mat. (e.g. Eq. S5), or after seeing Fig. 4, that this becomes more unambiguous.*

We agree with the reviewer and thank them for pointing this out. The following text is now included in the main text:

“Here the magnetic field B is offset at a small angle $\theta = 17^\circ$ from B_{SO} in the x-y plane (see Fig 2a).”

3) *In Suppl. Mat., the authors pay attention to the potential along x. But what about the potential in the y-z plane, so normal to the wire? Specifically, what is the depletion layer width at the wire boundaries? Is it of the same order as λ ? Is the remaining effectively conducting width or diameter of the wire of the order of $\frac{1}{2}$ the Fermi wavelength, so that it is compatible with QPC operation?*

In the simulations, we do take into account the full 3D structure of the nanowire device: The potential in the wire is obtained from the 3D FEM grid depicted in Fig. S1a. Since computing the quantum transport in 3D structure is computationally expensive we have then projected the 3D potential into an effective 1D potential (along the x-direction) using the transverse wave function of the nanowire $\psi(y,z)$. This is justified due to the large level spacing to the next higher modes.

Our 1D calculation thus implicitly also take into account the transverse wave function in the nanowire and the transverse confinement is indeed compatible with a QPC .

4) *The energy band picture inset in Fig. 1b is not explained and not used. It is potentially useful however. The chemical potential μ has the same symbol as the magnetic moment due to a typo in the latter (italic on line 3 of the manuscript, straight on line 4).*

We thank the reviewer for pointing this out and apologize for the typo. This has been corrected.

An explanation of the energy bands in Fig 1b is now included in the Figure caption,

“The QPC channel of length L is defined by the two contacts. Underneath the nanowire contacts, many sub-bands are occupied as the contacts screen the gate electric field. In the nanowire channel away from the contacts, the chemical potential in the wire, μ , is tuned with V_g .”

5) *Typo in “Strength”, Suppl. Mat. contents table, VII.*

We thank the reviewer for pointing this out and apologize for the typo. This has been corrected.

REVIEWERS' COMMENTS:

Reviewer #2 (Remarks to the Author):

After detailed and careful comments provided by authors and numerous changes made to manuscript I recommend publication of this work.

Reviewer #3 (Remarks to the Author):

The authors have satisfactorily answered the questions that came up in the previous review, and they have altered the manuscript and supplementary materials accordingly. The work is now more complete and still sufficiently circumspect. It appears the work is ready for publication.